# Circulating Cell-Free DNA in Renal Cell Carcinoma: The New Era of Precision Medicine

**DOI:** 10.3390/cancers14184359

**Published:** 2022-09-07

**Authors:** Edoardo Francini, Giuseppe Nicolò Fanelli, Filippo Pederzoli, Sandor Spisak, Erika Minonne, Massimiliano Raffo, Hubert Pakula, Viktoria Tisza, Cristian Scatena, Antonio Giuseppe Naccarato, Massimo Loda, Pier Vitale Nuzzo

**Affiliations:** 1Department of Experimental and Clinical Medicine, University of Florence, 50134 Florence, Italy; 2Department of Pathology and Laboratory Medicine, Weill Cornell Medicine, New York, NY 10021, USA; 3Division of Pathology, Department of Translational Research, New Technologies in Medicine and Surgery, University of Pisa, 56126 Pisa, Italy; 4Institute of Enzymology, Research Center for Natural Sciences, 1117 Budapest, Hungary; 5School of Medicine and Surgery, University of Turin, 10124 Turin, Italy; 6Unit of Urology, Division of Experimental Oncology, University Vita-Salute San Raffaele, URI, IRCCS Ospedale San Raffaele, 20132 Milan, Italy

**Keywords:** kidney cancer, liquid biopsy, cell-free methylated DNA, genetic, epigenetic, microsatellite instability

## Abstract

**Simple Summary:**

Early diagnosis of renal cell carcinoma (RCC) is challenging and typically incidental. Currently, several therapeutic strategies are used for the treatment; however, no established predictive biomarker has been established yet, and the optimal treatment choice and sequence of use remain unclear. Moreover, the recurrence occurs in about one-third of patients after tumor resection. Although several prognostic classification systems have been proposed, most of them showed only limited potential in recurrence prediction. Therefore, identifying simple, reliable, and easily accessible biomarkers to anticipate the diagnosis, effectively evaluate the risk of relapse, and predict the response to the therapeutic regimens is an unmet clinical need. Circulating cell-free DNA (cfDNA), released from cancer cells into the bloodstream, was shown to be a non-invasive, viable, inexpensive method to diagnose and monitor several solid malignancies, designed as a potential blood RCC biomarker. This review aims to summarize the state of the art of the current genetic and epigenetic techniques of plasma and serum cfDNA detection and outline the potential application of liquid biopsy in RCC.

**Abstract:**

Tumor biopsy is still the gold standard for diagnosing and prognosis renal cell carcinoma (RCC). However, its invasiveness, costs, and inability to accurately picture tumor heterogeneity represent major limitations to this procedure. Analysis of circulating cell-free DNA (cfDNA) is a non-invasive cost-effective technique that has the potential to ease cancer detection and prognosis. In particular, a growing body of evidence suggests that cfDNA could be a complementary tool to identify and prognosticate RCC while providing contemporary mutational profiling of the tumor. Further, recent research highlighted the role of cfDNA methylation profiling as a novel method for cancer detection and tissue-origin identification. This review synthesizes current knowledge on the diagnostic, prognostic, and predictive applications of cfDNA in RCC, with a specific focus on the potential role of cell-free methylated DNA (cfMeDNA).

## 1. Introduction

Kidney cancer is the third most frequent urologic cancer in adults and accounts for approximately 2–3% of all annually diagnosed malignancies in the United States [1]. Renal cell carcinoma (RCC) accounts for the vast majority of kidney cancers. RCC early diagnosis is challenging and typically incidental, primarily relying on radiological procedures executed for other indications [2]. Radical or partial nephrectomy offers curative responses in patients with localized RCC and selected patients with advanced RCC [3]. However, the disease recurs in about one-third of patients after resection of the primary tumor [4]. Existing prognostic classification systems based on clinical-pathological features, such as TNM stage and Fuhrman grading, showed limited potential to predict recurrence [5]. Currently, several therapeutic modalities comprising tyrosine kinase inhibitors, mammalian target rapamycin inhibitors, and immunotherapeutic agents are used for the treatment of metastatic RCC (mRCC) [6,7,8]. However, no established biomarker predictive of response to any of these treatments has been established yet, and there is uncertainty regarding their optimal choice and sequence of use in clinical practice. Therefore, identifying simple, accurate, and easily accessible biomarkers to boost the diagnostic specificity, effectively evaluate the risk of relapse following nephrectomy and predict the response to the therapeutic regimens currently offered for metastatic disease is an unmet clinical need.

To date, molecular hallmarks, assessed directly in primary tumor tissue, drive therapy and predict prognosis in several types of solid malignancies [9,10,11,12,13,14,15,16,17]; however, often, their evaluation is not reliable and presents some limitations [18,19,20,21]. Circulating tumor DNA (ctDNA), DNA released from cancer cells into the bloodstream or other body fluids, has been increasingly investigated as a “liquid biopsy” and proved to enable comprehensive genomic profiling of several tumors at various time points [22,23,24,25,26,27,28,29]. Further, the analysis of cfDNA was shown to be a viable, inexpensive, and less-invasive method to diagnose and monitor cancer hence circumventing the well-known shortcomings of invasive tissue biopsies [26,30]. In the recent past, as the concordance of genetic alterations between cfDNA and matched tumor biopsy was validated, there has been growing enthusiasm for the use of cfDNA as a potential blood cancer biomarker. In addition, epigenetic features of cfDNA, such as methylation of cfDNA (cfMeDNA), have been increasingly investigated in recent years [31,32]. Methylation of the pyrimidine ring of cytosine was shown to inhibit transcription when found in CpG islands and stabilize the genome when found in non-coding DNA regions [32,33]. The stability and specificity of CpG island methylation patterns demonstrated high sensitivity for detecting and classifying several tumor types, and thus, the analysis of DNA methylome is a promising cancer biomarker [34,35,36].

This review aims to describe the current genetic and epigenetic techniques of plasma and serum cfDNA detection and outline the potential application of cfDNA as a liquid biopsy in RCC, emphasizing its role in cancer detection, recurrence, prognosis, and prediction of therapy outcomes.

## 2. Data Acquisition

In this non-systematic literature review, PUBMED and MEDLINE were thoroughly searched through 20 April 2022 for peer-reviewed prospective and retrospective studies, reviews, and abstracts regarding cfDNA and cfMeDNA applied to detect, monitor, or prognosticate RCC. To this aim, keywords used included “kidney neoplasm”, “renal cell cancer”, “cell-free DNA”, “circulating DNA”, “circulating tumor DNA”, “methylated DNA”, and “liquid biopsy”. The documents more relevant to this review were selected, reviewed, and their data extracted.

## 3. Isolation and Analysis of cfDNA

The efficiency and consistency of the cfDNA extraction process are impacted by the technique chosen to isolate cfDNA, which is also partly responsible for the significant heterogeneity of the cfDNA data among the studies reported in this review. The key technologies used to obtain cfDNA from blood can be divided into “commercial methods” including anion-exchange-based ready-to-use extraction package, silica-membrane, or magnetic-particle technique, and “non-commercial methods,” such as the extraction of phenol-chloroform, guanidine-resin, alcohol precipitation, and salting-out processes.

Only two studies in the current review performed cfDNA extraction with phenol-chloroform extraction and ethanol precipitation in RCC patients [37,38]. Although many “non-commercial methods” achieve high yields and allow for the extraction of more small-sized fragments, they include toxic solvents, are more time-consuming, and are generally not considered appropriate for clinical analyses due to the high coefficient of results variability. Therefore, most studies of cfDNA are performed using “commercial methods”. They have good repeatability and reproducibility and thus enable standardization for clinical studies, although the loss of small and informative cfDNA fragments during the isolation process results in poor recovery rates. Therefore, the development of novel techniques is urgently required to improve the efficiency and quality of cfDNA extraction. cfDNA extraction techniques are summarized in Figure 1.

Once cfDNA is isolated, different approaches can be used for analysis. Ideally, cfDNA analytical techniques should work with low and very low inputs of genomic material, have a high discriminatory capacity to distinguish between cancer ctDNA and germline cfDNA shed by normal cells, show a reasonable cost-benefit ratio, and be reproducible among centers and in longitudinal cohorts. Two different strategies can be implemented: targeted approaches, which allow the detection of few, *a priori* selected genes, or untargeted approaches, which perform a genome-wide analysis of potential genomic alterations and do not require a selection of genes of interest.

Targeted approaches encompass a variety of traditional candidate gene methods (e.g., real-time polymerase chain reaction (real-time PCR), droplet digital PCR (ddPCR), and beads, emulsions, amplification and magnetics (BEAMing) assay) and next-generation sequencing (NGS)-based techniques (e.g., AmpliSeq, SAFE-SeqS, CAPP-Seq, and MCTA-Seq). Real-time PCR offers a cheap and easy-to-setup platform to study targeted genes with a very low rate of false positives and with no need for complex bioinformatic pipelines to analyze the data. Different protocols have been developed to increase sensitivity, including allele-specific amplification (AS-PCR) [39] and co-amplification at lower denaturation temperature (COLD-PCR) [40]. Of note, AS-PCR protocols are already clinically implemented to study small genomic alterations, such as indels or single nucleotide variations. Therefore, the implementation of these approaches to ctDNA analysis may expand the testing capacity at many clinical sites. The two techniques based on digital PCR—ddPCR and BEAMing—allow the detection of known point mutations present at allele frequencies of <0.01% [41,42]. Both techniques are based on the idea of generating DNA-containing droplets to be used to detect and quantify the presence of a particular genomic aberration [43,44]. Indeed, several studies have proven the utility of digital PCR-based approaches in different solid malignancies, including breast, lung, and colorectal cancer [45,46,47,48]. Despite optimal rates of specificity, sensitivity, and concordance between genomic alterations found in the primary tumor and in the ctDNA, these techniques are limited by the low number of *a priori* selected alterations that can be tested for reaction, as well as a complicated and expensive workflow for the BEAMing technique. NGS-based targeted approaches rely on the ability to sequence in parallel millions of DNA sequences to identify potential genetic alterations. Several methods have been designed to detect known copy number alterations and point mutations, overcoming at the same time the random error rate intrinsic to NGS approaches [49]. A thorough discussion of the specific technical details of each NGS-based approach is beyond the scope of this review. In general, while these methods allow for interrogation of a higher number of known alterations per run compared to the previously discussed approaches, they require bioinformatics training and are generally more laborious and expensive. At the same time, rigorous benchmarking of several analytic techniques is needed to determine the best pipelines for ctDNA analyses. Finally, as targeted approaches allow for accurate detection of selected genomic alterations, they do not address well tumor clonal heterogeneity and new mutations in response to therapy.

On the other hand, untargeted, NGS-based profiling of ctDNA by deep sequencing offers the opportunity to perform a genome-wide analysis of copy number alterations and mutations [50]. Moreover, this approach allows the discovery of novel alterations that were not present in the primary tumor or in the *a priori* selected panel of candidate genes, including de novo alterations arising during tumor progression or therapy. Untargeted sequencing appears particularly appealing, especially in the context of precision oncology protocols, where the emergence of a novel genomic alteration characterized through ctDNA may explain therapeutic resistance or lead to the administration of targeted therapies against the detected alteration [51]. However, the routine clinical application of this approach is still hampered by higher costs and lower specificity (∼80–90%) compared to other methods [49], even if the introduction of molecular barcoding or identifiers generally resulted in lower rates of false negatives [52,53]. cfDNA analysis approaches are summarized in Figure 1.

While every single approach has advantages and disadvantages, the wide implementation of ctDNA liquid biopsy in the future will be based on a combination of different targeted and untargeted techniques. Targeted methods would be potentially used to detect the presence of well-described driver genomic alteration with great confidence, while untargeted sequencing would be used longitudinally to monitor tumor clonal plasticity, the emergence of therapy resistance, or novel, potentially druggable mutations.

## 4. Applications of cfDNA Analysis in RCC

Recent studies suggest that the technological advancements in cfDNA assessment allow for the use of cfDNA as a reliable biomarker of cancer diagnosis, potentially permitting an early identification and treatment of RCC [54]. The evaluation of cfDNA levels, integrity, and genetic and epigenetic alterations showed to be useful for monitoring post-operative recurrence, predicting response to targeted therapies, and indicating the prognosis of RCC. The detailed applications of cfDNA in the detection, monitoring, and prognostication of RCC are described below.

### 4.1. Diagnostic Role

RCC is typically detected by an abdominal ultrasound examination [2]. However, owing to its precision and accuracy limitations, suspicious findings must be validated by CT or MRI scans [55]. While the analysis of cfDNA could aid the detection of early signs of the disease and the distinction between benign and malignant renal lesions, there are no clear recommendations yet regarding the use of cfDNA for these aims. A major hurdle to RCC diagnosis by means of cfDNA analysis is that of all extracranial tumors studied, RCC sheds the least amount of ctDNA in the bloodstream, and its detection is challenging [56,57,58]. In addition, the relevant discrepancies of cfDNA isolation and detection techniques among the studies investigating cfDNA in RCC make comparisons of findings quite challenging. Standardization of these methods within large prospective trials is warranted to validate cfDNA as a screening biomarker for RCC.

#### 4.1.1. cfDNA Levels in RCC, Healthy Controls and Non-Cancer Disease

Five prospective studies evaluated cfDNA levels in a total of more than 300 RCC patients and 140 control subjects [57,59,60,61,62]. Although an analysis of the aggregate data of these reports is biased by the several heterogeneities among the studies, including cfDNA sources (serum versus plasma), isolation and evaluation techniques, and populations with different disease states, it should be noted that in all five analyses the levels of cfDNA in RCC patients were higher than in healthy controls.

The analysis by Perego et al. of 48 patients with RCC and 41 healthy subjects showed that the mean pre-operative plasma cfDNA concentration in the RCC group was eight times higher than that of the healthy controls (26.4 vs. 3.2 ng/mL, *p* = 0.003) and decreased after nephrectomy [59]. Of note, the significance with respect to the controls persisted even when patients were stratified by sex, age, histology, tumor size, grading, and pathological TNM stratification [59].

The largest study assessing cfDNA levels conducted so far, including 157 patients with RCC and 43 with benign lesions, showed increased pre-operative cfDNA levels in those with the malignancy than in those with benign tumors (3319 vs. 1288 genetic equivalents/mL, *p* < 0.001), implying that cfDNA may help in the differential diagnosis of a solid renal masses [60]. Furthermore, total cfDNA levels were associated with tumor necrosis, lymph node involvement, and metastatic disease [60]. Wan et al. documented increased levels of cfDNA in patients with metastatic RCC (mRCC) compared to patients with localized RCC and healthy individuals [57]. Further, in this case, they found a significant correlation of plasma cfDNA levels with different Fuhrman grades, TNM stages, and tumor sizes, suggesting an association with tumor aggressiveness [57]. This seems to find confirmed in the study by Feng et al., where cfDNA levels were significantly associated with TNM stage, Fuhrman grade and number of metastases increased pretreatment amounts of cfDNA were similarly found in mRCC patients compared to healthy individuals [61].

Although, in aggregate, these data seem to suggest a correlation between raised plasma or serum levels of cfDNA and the presence of RCC, the cumulative body of evidence is still not sufficiently solid. Because increased cfDNA may be found in other non-cancer conditions, such as benign lesions, inflammatory or autoimmune diseases, and tissue traumas, the utility of cfDNA quantification to detect the presence of RCC is indeed limited [63,64,65]. Therefore, cfDNA analysis alone should not be considered a valid screening instrument for RCC at present.

#### 4.1.2. cfDNA Integrity as a Tool for Differentiating RCC and Non-Cancer Controls

Postulating that cfDNA derived from cancer was released mainly by necrosis compared to the cfDNA released in non-malignant diseases which origins mainly by apoptosis, three studies assessed size variations of cfDNA fragments to determine potential associations with the presence of RCC [66,67,68].

Using quantitative real-time PCR, Hauser et al. analyzed the serum cfDNA integrity, defined as the ratio of the longer fragment actine-beta gene 384 (ACTB^384^) derived from necrosis and the shorter fragment actine-beta gene 106 (ACTB^106^) derived from apoptosis of 35 RCC patients and 54 healthy controls [66]. The levels of both cfDNA fragments were found to increase in the pre-operative serum of RCC patients compared to the healthy individuals, indicating that cfDNA is fragmented to a higher degree in cancer patients [66]. Further, the significantly higher quantity of ACTB^384^ compared to ACTB^106^ observed in RCC supported the hypothesis of the necrotic origin of the cfDNA derived from the tumor. Consistently with these findings, Gang et al. performed conventional PCR using different primers able to detect the different sizes of the product of the housekeeping gene glyceraldehyde-3-phosphate dehydrogenase gene (GADPH) and observed a higher concentration of long fragments in the serum of pre-operative RCC patients compared to those in the healthy control group, indicating that necrosis is a more common occurrence in RCC [67]. Moreover, in this cohort, cfDNA integrity was also significantly associated with tumor stage and size [67].

In contrast, a study by Lu et al. found no difference in the quantity of the cfDNA fragments among patients with mRCC, non-metastatic RCC, and healthy controls [68]. However, it should be noted that this analysis was performed on plasma as opposed to serum cfDNA. Moreover, in this study, there was a decrease in cfDNA integrity in controls compared to those with metastatic disease, confirming that a higher cfDNA fragmentation could be indicative of RCC [68].

Although the findings of most studies support the hypothesis that ctDNA is more likely released by necrotic cancer cells and thus it is composed of larger fragments compared to non-tumor-derived cfDNA, likely released by apoptotic cells, the limited number of samples analyzed, the lack of standardization in measuring long vs. short fragments of cfDNA and the modest sensitivity and specificity values for short and long fragment detection prevent us from considering cfDNA size as a reliable diagnostic biomarker. The specific size of tumor-derived DNA fragments, the detailed characterization of tumor-derived alterations, and the potential implications of these biological differences have not been sufficiently explored yet.

#### 4.1.3. cfDNA Genetic Alterations

The identification of genetic alterations (GA), such as microsatellite instability (MSI) or loss of heterozygosity (LOH) in the cfDNA, is another feature of cfDNA which holds potential as a biomarker of early diagnosis of RCC. Diagnostic data of LOH and MSI in cfDNA from RCC are summarized in Table 1.

As mentioned above, some studies showed that, out of all extracranial tumors, RCC is one of the least producers of cfDNA, which results in a relatively low detection rate of cfDNA [56,57,58]. In turn, that seems to impair the probability of detecting GAs in cfDNA of patients with RCC. Indeed, GAs were detected in a percentage of patients with RCC ranging from 3.9% to 78.6% in analyses of cfDNA using panels of selected genes [28,58,69,70,71,72,73,74,75,76,77,78]. The GA identification rate was higher in those reports that used panels with a greater number of selected genes. Therefore, the observed high variability in GAs detection rate may partly depend on the number of genes investigated.

Goessl et al. detected GAs in plasma cfDNA of RCC patients by applying only four markers for microsatellite alterations (MSI or LOH) on chromosome 3p [79]. The observed alterations were not correlated with the stage of the tumor and were not found in the healthy controls [79]. Notably, although 63% of patients in this study had LOH in at least one locus and 35% in more than one locus, only one patient was found with MSI [79].

A similar rate of microsatellite alterations was observed in the serum-derived cfDNA of RCC patients in a smaller prospective study [37]. In this case, 60% of patients with malignant renal tumors had at least one alteration in the 28 microsatellite markers of the 20 chromosome regions investigated [37]. More importantly, this study also reported no cfDNA aberrations in healthy controls and no association between cfDNA alterations and tumor stage in those with RCC [37].

A later analysis focused on 20 microsatellite markers located on chromosomes 3p and 5q [80]. Serum cfDNA GAs were detected in 74% of cases using 9 markers, in 87% of cases using 11 markers, and only in 15% of controls using 10 markers, suggesting that cfDNA GAs may be associated with RCC [80]. In this respect, Perego et al. investigated five microsatellite alterations on chromosome 3p in nine of the 54 RCC patients whose pre-operative plasma was available [59]. Five of those nine patients had at least one of the five microsatellite markers, and these variations were also demonstrated in their primary tumor [59]. Although in aggregate, these data would portend a potential association between cfDNA GAs and RCC detection, the sensitivity and specificity of these analyses seem to depend on the number of microsatellites investigated. In this regard, the comprehensive study of microsatellites and GAs currently offered by NGS may be a more efficient and accurate method of analysis.

**Table 1 cancers-14-04359-t001:** Diagnostic data of microsatellite alterations (loss of heterozygosity [LOH] and/or microsatellite instability [MSI]) in circulating cell-free DNA (cfDNA) from renal cell carcinoma (RCC). AML, angiomyolipoma; cfDNA, cell-free DNA; MN, metanephricnephroma; NR, not reported; OCT oncocytoma; RCC, renal cell carcinoma; TCC, transitional cell carcinoma; MA, microsatellite alteration (loss of heterozygosity [LOH] and/or microsatellite instability [MSI]). * MA was performed only in 9 patients whose pre-operative plasma DNA was available.

Author, Year, [Ref.]	Microsatellite Markers	Patients (*n*)	Controls (*n*)	Sensitivity (%)	Specificity (%)
**Goessl, 1998, [79]**	D3S1307(3p), D3SI560(3p), D3SI289(3p), D3SI300(3p)	40 RCC	10 healthy individuals	63 (at least one MA) 35 (more than one MA)	100
**Eisenberger, 1999, [37]**	D1S251 (1pq), HTPO(2p), D3S1317(3p), D3S587(3p), D3S1560(3p), D3S1289(3p), D3S1286 (3p), D3S1038(3p), D4S243(4pq), FGA(4)(4q), CSF(5q), ACTBP2(5p), D8S348(8q), D8S307(8p), D9S747(9p), D9S242(9p), IFNa(9p), D9S162(9p), D11S488(11q), THO(11p), vWA(12p), D13S802(13q), MJD(14q), D17S695(17p), D17S654(17p), D18S51(18q), MBP(18q), D21S1245(21q).	25 RCC 1 AML 1 MN 3 OCT	8 individuals with nephrolithiasis 8 healthy individuals	60 (at least one MA)	100
**von Knobloch, 2002, [80]**	D3S1560(3p), D3S2450(3p), D3S3666(3p), D3S2408(3p), D3S1259(3p), D5S1720(5p), D5S1480(5p), D5S476(5p), D5S818(5p), D7S1796(7p), D7S1807(7p), D8S261(8p), D8S560(8p), D9S925(9p), D13S153(13p), D17S799(17p), D17S1306 (17p), D17S783(17p), D17S1298(17p), D17S807(17p)	53 RCC 1 renal B cell lymphoma 6 TCC	20 healthy individuals	74 (using 9 MA) 87 (using 20 MA)	85
**Perego, 2008, [59]**	D3S1566(3p), D3S1285(3p), D3S1300(3p), D3S1289(3p), D3S1597(3p)	48 RCC 1 TCC 5 OCT	41 healthy individuals	55.6 (at least one MA) *	NR

#### 4.1.4. cfDNA Epigenetic Alterations: A New Promising Method to Detect RCC

Another feature recently increasingly investigated as a potential diagnostic biomarker of RCC is the hyper- or hypo-methylation of cfDNA. Targeted methylation and global methylation analyses seemed to provide high specificity and sensitivity in distinguishing RCC patients from healthy controls compared to genetic analysis. The high tissue and cell type specificity of DNA methylation, as well as its high stability, would make it an ideal biomarker of RCC detection [31,32]. Diagnostic data of methylation analysis of cfDNA in patients with RCC are summarized in Table 2. Moreover, methylation of genes occurs frequently in the early stages of RCC [81]. For instance, methylation of the Ras association domain family member 1A (RASSF1A) seems to be fairly common in patients with RCC [38,60,62,82]. In this respect, recent analyses showed RASSF1A methylation ranging from 23% to 63% in RCC patients [38,60,62,82]. These discrepancies in the methylation frequency of the RASSF1A gene could be partly explained by the use of a bisulfate-based assay in lieu of the methylation-sensitive restriction enzyme assay, which is more sensitive.

The methylation of other genes with a key role in renal carcinogenesis also seems to hold a promising role in detecting RCC. In Hoque’s cohort, 67% of RCC patients had a methylated promoter in at least one of the nine genes analyzed [38]. cfDNA analysis in eight selected methylated genes (APC, GSTP1, p14(ARF), p16, RAR-B, RASSF1A, TIMP3, PTGS2) showed that 86% of the patients harbored at least one methylated gene [82]. Although all genes, except p16 and TIMP3, were significantly methylated in RCC patients compared to healthy individuals, only APC was correlated with the advanced tumor stage [82]. Furthermore, hypermethylation of the suppressor genes LRRC3B, APC, FHIT, and RASSF1 was detected in the plasma of RCC patients as opposed to healthy individuals; however, no correlation with clinicopathologic features was found [62].

These studies are limited by the use of PCR-based methods, which, albeit inexpensive and easy to perform, analyze only a limited number of methylated sites. An alternative and novel approach to overcome this issue is the analysis of the whole cfDNA methylome.

Whole genome bisulfite sequencing (WGBS), interrogating genome-wide methylation patterns, was recently performed in a large cohort of more than 2000 patients with different types of cancer, including 81 RCC across all stages [83]. The authors applied a classifier built on the differentially methylated regions (DMRs) between the cancers and controls in a training cohort and a validation cohort, reaching a specificity of 99.8% and 99.3%, respectively [83]. In addition, cfDNA detection showed increased sensitivity with increasing stages of the disease [83]. Specifically, in a pre-specified group of 12 cancer types, sensitivity raised from 39% (CI: 27% to 52%) in stage I, to 69% (CI: 56% to 80%) in stage II, 83% (CI: 75% to 90%) in stage III, and 92% (CI: 86% to 96%) in stage IV. This finding indicates the potential of cfDNA methylome not only to distinguish between benign or malign tumors but also to stage RCC.

Superior sensitivity and accuracy were achieved with cell-free methylated DNA immunoprecipitation and high-throughput sequencing (cfMeDIP-seq) [28,34,84,85]. CfMeDIP-seq is a sensitive, low-input (<10 ng of cfDNA), cost-efficient, and bisulfite-free technology allowing for the profiling of cfDNA methylomes. This approach enriches methylated DNA fragments in cfDNA by immunoprecipitation for high-throughput sequencing [34,84]. cfMeDIP-seq has been previously demonstrated to provide a high level of sensitivity for both the detection and classification of a plethora of localized and metastatic tumor types, including RCC [27,28,34,85,86,87]. The first independent validation of plasma cfMeDIP-seq was reported by Nuzzo et al. in a retrospective study on plasma cfDNA samples from 69 stage I–IV RCC patients and 13 normal controls [85]. The authors applied the classifier built on the top 300 DMRs between the RCC and controls, reaching a mean area under the receiver operating characteristic (AUC) curve of 0.990 [85]. Although they included RCC patients with clear cell and papillary histology, no difference in methylation were found between the histology, mainly because of the small number of patients with papillary histology. The same classifier was tested in an independent cohort of 34 patients with mRCC showing that cfMeDIPseq was able to identify all mRCC cases with 88% specificity and 100% detection rate [28].

A complete list of the studies that highlight the diagnostic role of cfDNA in RCC is reported in Table 3.

### 4.2. Post-Operative Recurrence and Prognostic Role

Earlier detection of metastatic disease may improve clinical outcomes. In this scenario, some reports indicated that cfDNA could have potential as a surveillance biomarker for patients with localized RCC after radical or partial nephrectomy, and few reports studied the association with the OS [57,59,60,74,77,88].

An intriguing analysis of pre-operative serum cfDNA of patients with clinically organ-confined RCC reported that all of those with two or three detectable LOH in the 28 studied loci experienced disease recurrence within 2 years after surgery vs. only 7% with no detectable serum LOH [88]. This finding suggested that microsatellite aberrations may detect post-surgery RCC relapse [88]. In contrast, another study analyzing plasma cfDNA of a smaller cohort of localized RCC patients showed that, although cfDNA concentration drastically decreased in all subjects after radical surgery compared to pre-nephrectomy levels, only 20% of those who showed an increase in cfDNA levels during the follow-up experienced recurrence [59]. Additionally, relapse-free survival was not associated with the presence of LOH in cfDNA [59]. A larger analysis reported the efficacy of plasma cfDNA in monitoring post-nephrectomy recurrence in a group of 92 RCC patients [57]. Patients with higher cfDNA levels had a significantly greater recurrence rate than those with lower levels before and after nephrectomy. Interestingly, two contemporary studies showed that RCC patients with detectable pre-operative cfDNA had shorter PFS compared to those without [74,77], suggesting that evidence of pre-surgery cfDNA levels may be a promising biomarker of recurrence in RCC patients following nephrectomy procedure. Similarly, a prospective study of 200 RCC patients with a median follow-up period of 28 months showed that higher pre-nephrectomy serum cfDNA levels were associated with shorter RCC-specific survival [60].

In a recent prospective study of RCC patients, a greater proportion of cfDNA fragments was associated with the presence of ctDNA, suggesting that those with ctDNA had shorter cfDNA fragment sizes. Poorer cancer-specific survival was significantly associated with the presence of ctDNA and shorter cfDNA fragment size [74,77]. These findings indicated that ctDNA status and cfDNA fragment size might be employed as biomarkers for prognosis in RCC.

Although the results of several studies suggest that identifying RCC molecular relapse before radiologic evidence and assessing cancer-specific survival by means of cfDNA analysis is a concrete possibility, solid data are still lacking. Larger prospective studies are needed to validate cfDNA as a biomarker for monitoring and predicting disease-specific survival.

Studies that validated the prognostic role of cfDNA in RCC are reported in Table 4.

### 4.3. Predictive Role

Few studies have explored the role of cfDNA in monitoring therapeutic efficacy in mRCC [61,69,70,89]. A report by Fang et al. on patients with mRCC showed a correlation between higher levels of plasma cfDNA and poor response to sorafenib [61]. Moreover, significantly lower levels of cfDNA were observed in those with remission or stable disease compared to those with disease progression [61]. In particular, the levels of cfDNA at 8 weeks of treatment were predictive of radiological progression [61]. In a similar fashion, a small prospective study of 23 mRCC patients treated with targeted therapy showed that pretreatment, higher levels of cfDNA were associated with shorter progression-free survival [89].

Recently, in the largest assessment to date of ctDNA of patients with mRCC, most subjects were found to have clinically relevant GAs [69]. The analysis of the ctDNA profile in a cohort of 220 mRCC patients and across patients receiving first and later-line therapies showed that, while the most common alterations were TP53, VHL, NF1, EGFR, and ARID1A, regardless of the line of treatment, the genetic profiles differed among patients receiving first-line compared to those receiving second-line treatments [69]. In particular, the authors observed GA rate differences in those who had second or later lines versus first-line therapy only, with higher variations found for TP53 (49% vs. 24%), VHL (29% vs. 18%), NF1 (20% vs. 3%), EGFR (15% vs. 8%), and PIK3CA (17% vs. 8%) [69]. Interestingly, these rate differences (particularly for TP53 and VHL) were even more evident when limiting the analysis to those receiving vascular endothelial growth factor inhibitors, which may indicate a mechanism of resistance to this type of treatment [69]. However, a much smaller cohort of 34 mRCC with serial plasma samples collected during therapy showed no association between cfDNA and response to treatment [70].

In aggregate, these data indicate that analysis of cfDNA in the course of the disease may have a predictive role in the treatment of mRCC, but larger datasets are warranted to confirm this hypothesis. Table 5 summarizes the studies that investigated the predictive role of cfDNA in RCC.

## 5. Conclusions

The use of cfDNA as a non-invasive biomarker for routine use in oncology has made great progress in recent years. As cfDNA is a fast, low-cost, and easy way to obtain dynamic information about the tumor, continual efforts from intense multidisciplinary studies have been made to transfer the research tools to routine clinical practice. On the one hand, several reports highlight cfDNA as a potential biomarker in RCC diagnosis, prognosis, and treatment and show how technological breakthroughs have brought new cancer detection methods with higher accuracy and efficiency. On the other hand, there are challenges to the widespread use of this new approach in the routine clinical setting. From a technical point of view, particular efforts have to be focused on the improvement of cfDNA technical assays, high-quality output, library preparation procedures, and sequencing depth for precise methylation analyses. Currently, most liquid biopsies tests are commonly used as a complementary technique to standard tissue biopsies, primarily in research settings, with the final aim of standardization of these techniques allowing their adoption in clinical routine.

Although the FDA has encouragingly approved some liquid biopsy tests, such as FoundationOne^®^Liquid CDx (Foundation Medicine, Inc.; Cambridge, MA, USA), in the next few years, the huge amount of data collected from the several ongoing prospective clinical trials based on the cfDNA as an integral biomarker will represent the key for further innovation in this promising field, corroborating the clinical evidence of clear benefits for patients, reducing medical overheads, increasing the early diagnostic options, and improving tailored treatments.

## Figures and Tables

**Figure 1 cancers-14-04359-f001:**
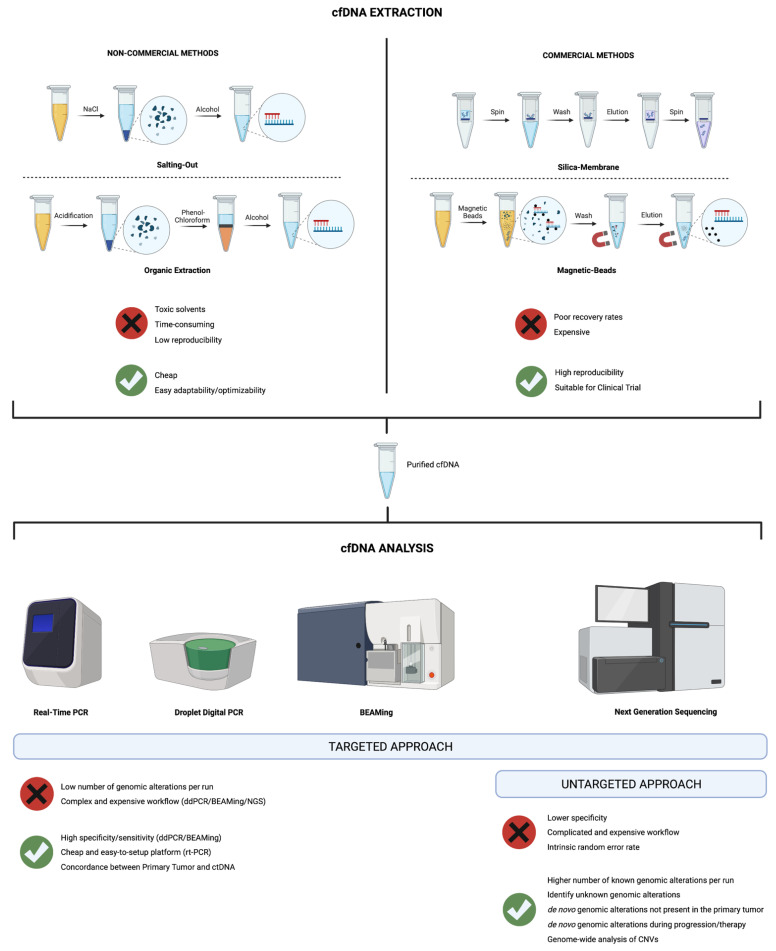
Graphical representation of the cfDNA extraction methods and analysis approaches. Advantages and disadvantages are highlighted for each technique.

**Table 2 cancers-14-04359-t002:** Diagnostic data of methylation analysis of circulating cell-free DNA (cfDNA) in patients with renal cell carcinoma (RCC). APC, adenomatosis-polyposis-coli gene; ARF, auxin response factors; CDH1, cadherin-1 gene; cfDNA, cell-free DNA; FHIT, fragile histidine triad gene; GSTP1, glutathione-S-Transferase Pi 1 gene; ITGA9, integrin subunit alpha 9 gene; LRRC3B, leucine rich repeat containing 3B gene; MGMT, O-6-methylguanine-DNA methyltransferase gene; NR, not reported; p16, cyclin-dependent kinase inhibitor 2A; PCR, Polymerase chain reaction; PTGS2, prostaglandin–endoperoxide.

Author, Year, [Ref.]	Gene	Methylated /RCC Patients (*n*), (%)	Methylated /Controls (*n*), (%)	Sensitivity (%)	Specificity (%)	Cutoff Value	AUC	Sample Source, Amount	Detection Method
**Hoque, 2004, [38]**	* **APC** *	1/18 (5.5)	1/30 (3.3)	5.5	96.7	4.5	NR	Serum, NR	PCR
	* **ARF** *	1/18 (5.5)	1/30 (3.3)	5.5	96.7	0	NR		
	* **CDH1** *	6/18 (33.3)	2/30 (6.6)	33.3	93.4	0.3	NR		
	* **GSTP1** *	1/18 (5.5)	0/30 (0)	5.5	100	0	NR		
	* **MGMT** *	0/18 (0)	1/30 (3.3)	0	96.7	0	NR		
	* **p16** *	4/18 (22.2)	0/30 (0)	22.2	100	0	NR		
	* **RAR-82** *	1/18 (5.5)	0/30 (0)	5.5	100	0.1	NR		
	* **RASSF1A** *	2/18 (11.1)	1/30 (3.3)	11.1	96.7	0.1	NR		
	* **TIMP3** *	3/18 (16.6)	0/30 (0)	17	100	1	NR		
**De Martino, 2011, [60]**	* **RASSF1A** *	72/157 (45.9)	3/43 (7)	45.9	93	0	0.694	Serum, 1 mL	qPCR
	* **PTGS2** *	60/157 (38.2)	15/43 (34.9)	38.2	65.1	0	0.517		
	* **P16** *	73/157 (46.5)	19/43 (44.2)	46.5	55.8	0	0.512		
	* **VHL** *	79/157 (50.3)	4/43 (8.3)	50.3	90.7	0	0.705		
**Hauser, 2013, [82]**	* **APC** *	19/35(54.3)	5/54 (9.3)	54.3	90.7	0.37	0.72	Serum, 1 mL	PCR
	* **GSTP1** *	6/35(17.1)	1/54(1.9)	17.1	98.1	0.75	0.57		
	* **p14(ARF)** *	5/35(14.3)	0/54(0)	14.3	100	0.26	0.57		
	* **P16** *	9/35(25.7)	9/54(16.7)	25.7	83.3	0	NR		
	* **PTGS2** *	8/35(22.9)	2/54(3.7)	22.9	96.3	0.47	0.59		
	* **RAR-B** *	14/35(40)	8/54(14.8)	40.0	85.2	0.19	0.61		
	* **RASSF1A** *	8/35(22.9)	1/54(1.9)	22.9	98.2	0.09	0.60		
	* **TIMP3** *	20/35(57.1)	21/54(38.9)	57.1	61.1	0	NR		
**Skrypkina, 2016, [62]**	** *APC* **	14/27 (51.9)	1/15(6.7)	51.9	93.3	0	NR	Plasma, 2 mL	PCR
	** *FHIT* **	15/27 (55.6)	0/15 (0)	55.6	100	0	NR		
	** *ITGA9* **	0/27(0)	0/15 (0)	0	100	0	NR		
	** *LRRC3B* **	20/27 (74.1)	5/15 (33.3)	74.1	66.7	0	NR		
	** *RASSF1* **	17/27 (63.0)	1/15 (6.7)	62.9	93.3	0	NR		
	** *VHL* **	0/27 (0)	0/15 (0)	0	100	0	NR		

**Table 3 cancers-14-04359-t003:** Summary of the studies evaluating the diagnostic role of circulating cell-free DNA (cfDNA) in renal cell carcinoma (RCC). AUC: area Under the Curve; p16:cyclin-dependent kinase inhibitor 2A; ACTB: actin beta gene; APC: adenomatosis-polyposis-coli gene; cfDNA: circulating cell-free DNA; CfMEDIP-seq: cell-free methylated DNA immunoprecipitation-sequencing; FHIT: fragile histidine triad gene: GA: genetic alteration; GSTP1: glutathione-S-Transferase Pi 1 gene; LOH: loss of heterozygosity; LRRC3B: leucine rich repeat containing 3B gene; mRCC: metastatic Renal Cell Carcinoma; NF1: neurofibromatosis type 1 gene; OS: overall survival; p14 ARF: auxin response factors p14; PCR: polymerase chain reaction; PFS: progression-free survival; PTGS2: prostaglandin–endoperoxide synthase; qPCR: real-time polymerase chain reaction; RAR-B: retinoic acid receptor beta; RASSF1: Ras association domain family member 1A; RCC: renal Cell Carcinoma; TIMP3: tissue inhibitor of metalloproteinase-gene; TP53: tumor protein p53; TOO: tissue of origin; VHL: von Hippel-Lindau; VAF: Variant allele frequency; WGBS: whole-genome bisulfite sequencing.

DIAGNOSTIC ROLE OF cfDNA IN RCC										
cfDNA Feature Analyzed	Author, Year, [Ref.]	Study Type	Purpose	Patients *(n)*	Controls	Sample Source, Amount	Timing Sample	cfDNA Isolation Method	Detection Method	Results
**cfDNA** **levels**	**Perego, 2008, [59]**	Prospective	Diagnostic Prognostic	48 RCC (stages I–IV), 1 TCC, 5 OCT	41 healthy individuals	Plasma, 1 mL	Pre-operative Post-operative	QIAamp DNA Mini kit (Qiagen)	qPCR	RCC patients had a higher plasma cfDNA concentration compared to the healthy controls. Plasma DNA concentration decreased after nephrectomy. During follow-up, plasma DNA increased in 12 patients without evidence of neoplasia: 3 patients successively relapsed. Pre-operative plasma DNA of 9 patients harbored LOH in 5 cases (55.6%). Augmented plasma DNA of 7 patients displayed LOH in 3 cases (42.9%) at follow-up, and in 1 case preceded the recurrence of the disease.
	**De Martino, 2012, [60]**	Prospective	Diagnostic Prognostic	157 RCC (stage I–IV)	43 benign renal tumors	Serum, 1 mL	Pre-operative	QIAampUltrasens Virus kit (Qiagen)	PCR	Total cfDNA levels and CpG island methylation of RASSF1A and VHL were highly diagnostic for RCC. cfDNA levels were associated with poorer DFS. RASSF1A and VHL methylation was not associated with DFS.
	**Feng, 2013, [61]**	Prospective	Diagnostic Predictive	18 RCC (stage IV)	10 healthy individuals	Plasma, 0.4 mL	Six timepoints during treatment with Sorafenib: before treatment, 4–8–12–16–24 weeks	QIAamp DNA Blood Mini Kit (Qiagen)	qPCR	Pretreatment cfDNA levels were increased in RCC vs. healthy individuals and were associated with stage, grade, and the number of metastases. cfDNA levels from weeks 8 to 24 of treatment were higher in those with disease progression than in those with stable disease or partial response. Levels of cfDNA at 8 weeks were predictive of progression.
	**Wan, 2013, [57]**	Prospective	Diagnostic Prognostic	92 RCC (stage I–IV)	44 healthy individuals	Plasma, 0.4 mL	Pre-operative Post-operative	QIAamp DNA Blood Mini Kit (Qiagen)	qPCR	Pretreatment levels of plasma cfDNA in pts with mRCC were significantly higher than in those with localized RCC or healthy individuals. cfDNA levels were associated with Fuhrman grade, TNM stage, and tumor size. Of pts with localized RCC, those with recurrence had a significantly higher plasma cfDNA level than those without. The pts with a high plasma cfDNA level had a significantly higher recurrence rate than those with a low plasma cfDNA level before and after nephrectomy.
	**Skrypkina, 2016, [62]**	Prospective	Diagnostic	27 RCC (stage I–IV)	15 healthy individuals	Plasma, 2 mL	Pre-operative	Proba Na kit (DNA-Technology)	qPCR	Levels of cfDNA levels were significantly higher in RCC pts than in controls. Hypermethylation of CpG islands of LRRC3B, APC, FHIT, and RASSF1 genes was detected in RCC and was not correlated with clinicopathologic features.
**cfDNA** **integrity**	**Hauser, 2010, [66]**	Prospective	Diagnostic Prognostic	35 RCC (stage I–IV)	54 healthy individuals	Serum, 1 mL	Pre-operative	ChargeSwitchgDNA Kit (Invitrogen)	qPCR	cfDNA integrity (ACTB-384/ACTB-106 ratio) analysis distinguished between RCC and healthy controls. High level of ACTB384 compared to ACTB106 was found in RCC patients. No significant correlation of cfDNA levels and DNA integrity with pT-stage, grading or histological subtype was observed.
	**Gang, 2010, [67]**	Prospective	Diagnostic Prognostic	78 RCC (stage I–III)	42 healthy individuals	Serum, 0.4 mL	Pre-operative Post-operative	QIAamp DNA Blood Mini Kit (Qiagen)	PCR	cfDNA integrity distinguished between RCC and healthy controls and was correlated with tumor stage and size.
	**Lu, 2016, [68]**	Retrospective	Diagnostic Prognostic	229 RCC (stage I-IV)	40 healthy individuals	Plasma, 1 mL	Pre-operative Pre-therapy	QIAamp Circulating Nucleic Acid Kit (Qiagen)	PCR	cfDNA integrity did not differ in metastatic, non-metastatic and controls, but decreased from controls to metastatic patients.
**cfDNA** **genetic** **alterations**	**Goessl, 1998, [79]**	Prospective	Diagnostic	40 RCC (stages I–IV)	10 healthy individuals	Plasma, 1 mL	Pre-operative	Qiamp Blood Kit (Qiagen)	PCR	Analysis of LOH and microsatellite instability of four chromosome 3p microsatellites: LOH was found in at least 1 locus in 63% of pts, and 35% exhibited LOH at more than one locus. Microsatellite instability of plasma cfDNA was detected in 3% of the patient. No alterations were found in the controls.
	**Eisenberger, 1999, [37]**	Prospective	Diagnostic	25 RCC (stages I–II–III), 1 AML, 1 MN, 3 OCT	8 individuals with nephrolithiasis 8 healthy individuals	Serum, NR Urine, NR	Pre-operative	Digestion methods with proteinase K (Boehringer Mannheim GmbH, Mannheim, Germany) in the presence of sodium dodecyl sulfate, followed by phenol–chloroform extraction and ethanol precipitation	PCR	Analysis of LOH and microsatellite instability of 28 microsatellites markers analysis: 60% of RCC had one or more microsatellite cfDNA alterations in the serum. No alterations were found in the controls.
	**von Knobloch, 2002, [80]**	Prospective	Diagnostic	53 RCC (stages I–II–III), 1 renal B cell lymphoma, 6 TCC	20 healthy individuals	Serum, 2–4 mL	Pre-operative	Qiamp Midi-Kit (Qiagen)	PCR	Analysis of microsatellite instability of 20 microsatellites markers analysis: serum cfDNA alterations was detected in 74% of cases using 9 markers and 15% of controls using 10 markers.
	**Perego, 2008, [59]**	Prospective	Diagnostic Prognostic	48 RCC (stages I–IV), 1 TCC, 5 OCT	41 healthy individuals	Plasma, 1 mL	Pre-operative Post-operative	QIAamp DNA Mini kit (Qiagen)	PCR	RCC pts had a higher plasma cfDNA concentration compared to the healthy controls. Plasma DNA concentration decreased after nephrectomy. During follow-up, plasma DNA increased in 12 patients without evidence of neoplasia and 3 patients successively relapsed. Pre-operative plasma DNA of 9 patients harbored LOH in 5 cases (55.6%). Augmented plasma DNA of 7 patients displayed LOH in 3 cases (42.9%) at follow-up, and in 1 case preceded the recurrence of disease.
	**Bettegowda, 2014, [58]**	Prospective	Diagnostic	5 RCC (stage IV)	-	Plasma, 2 mL	-	QIAamp Circulating Nucleic Acid Kit (Qiagen)	PCR	ctDNA from RCC showed the lowest levels compared to other cancers.
	**Pal, 2017, [69]**	Prospective	Diagnostic Predictive	220 RCC (stage IV)	-	Plasma, 1.5–2 mL	Pre-therapy Post-therapy	QIAamp Circulating Nucleic Acid Kit (Qiagen)	Targeted sequencing (73 genes)	GAs were detected in 78.6% of patients. GAs in TP53 and NF1 increased with subsequent therapies.
	**Maia, 2017, [70]**	Retrospective	Diagnostic Predictive	34 RCC (stage IV)	-	Plasma, 1.5–2 mL	Pre-therapy Post-therapy	QIAamp Circulating Nucleic Acid Kit (Qiagen)	Targeted sequencing (73 genes)	GAs were detected in 53% pf patients. cfDNA showed to be is a surrogate of tumor burden burden. No associations were found between IMDC risk, histology or treatment type and presence/absence of cfDNA
	**Hahn, 2017, [76]**	Prospective	Diagnostic	19 RCC (stage IV)	-	Plasma, 1.5–2 mL	Pre-operative	QIAamp Circulating Nucleic Acid Kit (Qiagen)	Targeted sequencing (73 genes)	The median GAs rate in cfDNA was detected 2.2% of patients. Median mutation rate was similar between cfDNA and tumor tissue whereas concordance rate was 8.6%
	**Corrò, 2017, [73]**	Prospective	Diagnostic	9 RCC (stage I–III)	-	Plasma, 1 mL Serum, 1 mL	Pre-operative	QIAamp Circulating free DNA Kit (Qiagen)	PCR	The VHL mutation in plasma or serum was detected in 1one patients
	**Mouliere, 2018, [78]**	Prospective	Diagnostic	33 RCC (stage not available)	65 healthy controls	Plasma, 2 mL	Pre-operative	QIAamp Circulating Nucleic Acid Kit (Qiagen)	Low-pass whole-genome sequencing	Enrichment of ctDNA in fragment sizes between 90 and 150 bp improved RCC detection
	**Yamamoto, 2019, [71]**	Prospective	Diagnostic Prognostic	53 RCC (stage III–IV)	-	Plasma, 1–3 mL	Pre-therapy Post-therapy	QIAamp Circulating Nucleic Acid Kit (Qiagen)	ddPCR and targeted sequencing (48 genes)	GAs were detected in 30% of patients. ctDNA status and cfDNA fragmentation were associated with PFS and OS. Patients with detectable ctDNA had poor responses to therapy
	**Lasseter, 2020, [28]**	Retrospective	Diagnostic	40 RCC (stage IV)	34 healthy controls	Plasma, 1 mL	Pre-therapy Post-therapy	QIAamp Circulating Nucleic Acid Kit (Qiagen)	Targeted sequencing (27 genes) and cfMeDIP-seq	Genetic variants were found in 21% of the patients. cfMeDIP-Seq performed in 34 RCC patientsdetected all RCC cases (sensitivity 100%, specificity 88%)
	**Zhang, 2020, [72]**	Prospective	Diagnostic	50 RCC (stage IV)	-	Plasma, NA	During therapy	QIAamp Circulating Nucleic Acid Kit (Qiagen)	Targeted sequencing (120 genes)	GAs were identified in all 50 patients. The number of GAs was significantly associated with the number of lines of therapy
	**Bacon, 2020, [77]**	Prospective	Diagnostic Prognostic	55 RCC (stage IV)	-	Plasma, 1.5–2 mL	Pre-therapy	QIAamp Circulating Nucleic Acid Kit (Qiagen)	Targeted sequencing (981 cancer genes)	ctDNA was detected in 33% of patients and the average VAF was 3.9% Patients with detectable ctDNA had shorter PFS and OS
	**Smith, 2020, [74]**	Prospective	Diagnostic	29 RCC (stage I–IV)	-	Plasma, 2 mL	Pre-operative Post-operative	QIAamp Circulating Nucleic Acid Kit (Qiagen)	Targeted sequencing (297 genes)	cfDNA levels in RCC were significantly lower that in other analyzed cancers of similar size and stage. Targeted sequecing methods detected 27.5% of RCC patients. Post-operative cfDNA was correlated with clinical response to treatment.
	**Wan, 2020, [75]**	Prospective	Diagnostic	24 RCC (stage I–II)	45 healthy controls	Plasma, 1–2 mL	Pre-operative Post-operative	QIAamp Circulating Nucleic Acid Kit (Qiagen)	Targeted sequencing	cfDNA from RCC had the lowest levels compared to other analyzed cancers. AUC of the method was 0.66
**cfDNA** **epigenetic** **alterations**	**Hoque, 2004, [38]**	Prospective	Diagnostic	18 RCC (stages I–IV)	30 healthy individuals	Serum, 1 mL Urine, NR	Pre-operative	Digestion methods with proteinase K (Boehringer Mannheim GmbH, Mannheim, Germany) in the presence of sodium dodecyl sulfate, followed by phenol–chloroform extraction and ethanol precipitation	qPCR	Aberrant methylation of nine gene promoters’ analysis: 67% of RCC pts and 1% of controls were methylation positive for at least one gene tested
	**De Martino, 2012, [60]**	Prospective	Diagnostic Prognostic	157 RCC (stage I–IV)	43 benign renal tumors	Serum, 1 mL	Pre-operative	QIAampUltrasens Virus kit (Qiagen)	PCR	cfDNA levels and CpG island methylation of RASSF1A and VHL were highly diagnostic for RCC. cfDNA levels were associated with poorer DFS. RASSF1A and VHL methylation was not associated with DFS.
	**Hauser, 2013, [82]**	Prospective	Diagnostic	35 RCC (stage I–IV)	54 healthy individuals	Serum, 1 mL	Pre-operative	ChargeSwitchgDNA Kit (Invitrogen)	PCR	cfDNA methylation analysis in eight selected genes (APC, GSTP1, p14(ARF), p16, RAR-B, RASSF1A, TIMP3, PTGS2): in 30 of 35 pts with RCC, at least one gene was methylated. All genes, except p16 and TIMP3, were significantly methylated in RCC pts compared to healthy individuals. APC was correlated with advanced tumor stage
	**Skrypkina, 2016, [62]**	Prospective	Diagnostic	27 RCC (stage I–IV)	15 healthy individuals	Plasma, 2 mL	Pre-operative	Proba Na kit (DNA-Technology)	qPCR	Levels of cfDNA were significantly higher in RCC patients than in controls. Hypermethylation of CpG islands of LRRC3B, APC, FHIT, and RASSF1 genes was detected in RCC and was not correlated with clinicopathologic features.
	**Liu, 2020, [83]**	Prospective	Diagnostic	56 RCC in the training cohort 25 RCC in the validation cohort	1521 healthy individuals in the training cohort 610 healthy individuals in the validation cohort	Plasma, up to 10 mL	Pre-operative	QIAamp Circulating Nucleic Acid Kit (Qiagen)	WGBS (103,456 regions identified)	WGBS from multiple cancer types was used to build a classifier to identify the cfDNA and the TOO. cfDNA from RCC has the lowest detection, but TOO was classified correctly from all RCC patients.
	**Lasseter, 2020, [28]**	Retrospective	Diagnostic	40 RCC (stage IV)	34 healthy controls	Plasma, 1 mL	Pre-therapy Post-therapy	QIAamp Circulating Nucleic Acid Kit (Qiagen)	Tumor sequencing (27 genes) and cfMeDIP-seq	Genetic variants were found in 21% of the patients. cfMeDIP-Seq perfomed in 34 RCC patients detected all RCC cases (sensitivity 100%, specificity 88%)
	**Nuzzo, 2020, [85]**	Retrospective	Diagnostic	99 RCC (stage I–IV)	28 healthy controls	Plasma, 1 mL	Pre-operative	QIAamp Circulating Nucleic Acid Kit (Qiagen)	cfMeDIP-seq	Based on a classifier built on top 300 differentially methylated regions, cfMeDIP-seq detected cfDNA RCC in 97% of the patients

**Table 4 cancers-14-04359-t004:** Summary of the studies evaluating the prognostic role of circulating cell-free DNA (cfDNA) in renal cell carcinoma (RCC). LOH: loss of heterozygosity; mRCC: metastatic Renal Cell Carcinoma; NF1: neurofibromatosis type 1 gene; OS: overall survival; p14 ARF: auxin response factors p14; PCR: polymerase chain reaction; PFS: progression-free survival; PTGS2: prostaglandin–endoperoxide synthase; qPCR: real-time polymerase chain reaction; RASSF1: Ras association domain family member 1A; VHL: von Hippel-Lindau.

PROGNOSTIC ROLE OF cfDNA IN RCC									
Author, Year	Study Type	Purpose	Patients *(n)*	Controls	Sample Source, Amount	Timing Sample	cfDNA Isolation Method	Detection Method	Results
**Gonzalgo, 2002, [88]**	Retrospective	Prognostic	25 RCC (stages I–II–III), 1 AML, 1 MN, 3 OCT	_	Serum, NR Urine, NR	Pre-operative	Digestion methods with proteinase K (Boehringer Mannheim GmbH, Mannheim, Germany) in the presence of sodium dodecyl sulfate, followed by phenol–chloroform extraction and ethanol precipitation	PCR	Analysis of 28 microsatellites markers in RCC patients who had recurrent disease 2 years after nephrectomy: recurrence disease was detected in 7% of patients with no detectable pre-operative serum LOH, 17% with 1 detectable serum LOH and 100% with 2 or 3 detectable serum LOH.
**Perego, 2008, [59]**	Prospective	Diagnostic Prognostic	48 RCC (stages I–IV), 1 TCC, 5 OCT	41 healthy individuals	Plasma, 1 mL	Pre-operative Post-operative	QIAamp DNA Mini kit (Qiagen)	qPCR	RCC pts had a higher plasma cfDNA concentration compared to the healthy controls. Plasma DNA concentration decreased after nephrectomy. During follow-up, plasma DNA increased in 12 patients without evidence of neoplasia: 3 patients successively relapsed. Pre-operative plasma DNA of 9 patients harbored LOH in 5 cases (55.6%). Augmented plasma DNA of 7 patients displayed LOH in 3 cases (42.9%) at follow-up, and in 1 case preceded the recurrence of disease.
**De Martino, 2012, [60]**	Prospective	Diagnostic Prognostic	157 RCC (stage I–IV)	43 benign renal tumors	Serum, 1 mL	Pre-operative	QIAampUltrasens Virus kit (Qiagen)	PCR	Total cfDNA levels and CpG island methylation of RASSF1A and VHL were highly diagnostic for RCC. cfDNA levels were associated with poorer DFS. RASSF1A and VHL methylation was not associated with DFS.
**Wan, 2013, [57]**	Prospective	Prognostic	92 RCC (stage I–IV)	44 healthy individuals	Plasma, 0.4 mL	Pre-operative Post-operative	QIAamp DNA Blood Mini Kit (Qiagen)	qPCR	Pretreatment levels of plasma cfDNA in pts with mRCC were significantly higher than in those with localized RCC or nealthy individuals. cfDNA levels were associated with Fuhrman grade, TNM stage, and tumor size. Of pts with localized RCC, those with recurrence had a significantly higher plasma cfDNA level than those without. The pts with a high plasma cfDNA level had a significantly higher recurrence rate than those with a low plasma cfDNA level before and after nephrectomy.
**Yamamoto, 2019, [71]**	Prospective	Diagnostic Prognostic	53 RCC (stage III–IV)	-	Plasma, 1–3 mL	Pre-therapy Post-therapy	QIAamp Circulating Nucleic Acid Kit (Qiagen)	ddPCR and targeted sequencing (48 genes)	GAs were detected in 30% of patients. cfDNA status and cfDNA fragmentation were associated with PFS and OS. Patients with detectable ctDNA had poor responses to therapy
**Bacon, 2020, [77]**	Prospective	Diagnostic Prognostic	55 RCC (stage IV)	-	Plasma, 1.5–2 mL	Pre-therapy	QIAamp Circulating Nucleic Acid Kit (Qiagen)	Targeted sequencing (981 cancer genes)	cfDNA was detected in 33% of patients and the average VAF was 3.9% Patients with detectable cfDNA had shorter PFS and OS
**Smith, 2020, [74]**	Prospective	Diagnostic	29 RCC (stage I–IV)	-	Plasma, 2 mL	Pre-operative Post-operative	QIAamp Circulating Nucleic Acid Kit (Qiagen)	Targeted sequencing (297 variants)	cfDNA levels in RCC were significantly lower than in other analyzed cancers of similar size and stage. Targeted sequecing methods detected 27.5% of RCC patients. Post-operative cfDNA was coreelated with clinical response to treatment.

**Table 5 cancers-14-04359-t005:** Summary of the studies evaluating the predictive role of circulating cell-free DNA (cfDNA) in renal cell carcinoma (RCC). AUC: area Under the Curve; gene: GA: genetic alteration; LOH: loss of heterozygosity; OS: overall survival; PCR: polymerase chain reaction; PFS: progression-free survival; qPCR: real-time polymerase chain reaction.

PREDICTIVE ROLE OF cfDNA IN RCC									
Author, Year	Study Type	Purpose	Patients *(n)*	Controls	Sample Source, Amount	Timing Sample	cfDNA Isolation Method	Detection Method	Results
**Feng, 2013, [61]**	Prospective	Diagnostic Predictive	18 RCC (stage IV)	10 healthy individuals	Plasma, 0.4 mL	Six timepoints during treatment with Sorafenib: before treatment, 4–8–12–16–24 weeks	QIAamp DNA Blood Mini Kit (Qiagen)	qPCR	Pretreatment cfDNA levels were increased in RCC vs. healthy individual and were associated with stage, grade, and number of metastases. cfDNA levels from weeks 8 to 24 of treatment were higher in those with disease progression than in those with stable disease or partial response. Levels of cfDNA at 8 weeks were predictive of progression.
**Rouvinov, 2017, [89]**	Prospective	Predictive	23 RCC (stage IV)	-	Serum, 1 mL	Six timepoints during treatment with targeted therapy: before treatment, 4–8–12–16–24 weeks	QIAamp Blood Kit (Qiagene)	qPCR	Patients with normal pretreatment cfDNA level had a better PFS versus patients with increased levels. In multivariate analysis, cfDNA levels was associated with PFS.
**Pal, 2017, [69]**	Prospective	Predictive	220 RCC (stage IV)	-	Plasma, 1.5–2 mL	Pre-therapy Post-therapy	QIAamp Circulating Nucleic Acid Kit (Qiagen)	Targeted sequencing (73 genes)	GAs were detected in 78.6% of patients. The number of GAs was not correlated to line of therapy, but GAs in TP53 and NF1 increased with subsequent therapies.
**Maia, 2017, [70]**	Retrospective	Predictive	34 RCC (stage IV)	-	Plasma, 1.5–2 mL	Pre-therapy Post-therapy	QIAamp Circulating Nucleic Acid Kit (Qiagen)	Targeted sequencing (73 genes)	GAs were detected in 53% pf patients. cfDNA is a surrogate of tumor burner. No associations were found between ctDNA and IMDC risk, histology and response to treatment.

## Data Availability

Not applicable.

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
