# Peer review of "Circulating Cell-Free DNA in Renal Cell Carcinoma: The New Era of Precision Medicine"

_cancers, 2022, doi:10.3390/cancers14184359_

Round 1

Reviewer 1 Report

Dear Editor, thank you so much for inviting me to revise this manuscript about renal cell carcinoma.

Liquid biopsy, based on the circulating tumor cells (CTCs) and cell-free nucleic acids has potential applications at multiple points throughout the natural course of cancer, from diagnosis to follow-up. The advantages of doing ctDNA assessment versus tissue-based genomic profile are the minimal procedural risk, the possibility to serial testing in order to monitor disease-relapse and response to therapy over time and to reduce hospitalization costs during the entire process. However, some critical issues related to ctDNA assays should be taken into consideration. The sensitivity of ctDNA assays depends on the assessment technique and genetic platforms used, on tumor-organ, stage, tumor heterogeneity, tumor clonality. The specificity is usually very high, whereas the concordance with tumor-based biopsy is generally low. In patients with renal cell carcinoma (RCC), qualitative analyses of ctDNA have been performed with interesting results regarding selective pressure from therapy, therapeutic resistance, exceptional treatment response to everolimus and mutations associated with aggressive behavior. Quantitative analyses showed variations of ccfDNA levels at different tumor stage. A plathora of recent studies has assessed this topic.

Thus, the paper addresses a current topic in this setting.

The manuscript is well written and organized. 

Figures and tables are comprehensive and clear.

The introduction explains in a clear and coherent manner the background of this study.

We suggest the following modifications:

·      Introduction section: although the authors correctly included important papers in this setting, we believe that the background of medical treatment for renal cell carcinoma should be further explained and some recently published studies should be cited within the introduction (PMID: 34894318;  PMID: 35193819), only for a matter of consistency. We think it might be useful to introduce the topic of this interesting paper.

·      Table 2 is too long. I would suggest to split it in different parts to help readability.

·      The authors should expand some sections, including a more personal perspective to reflect on. For example, they could answer the following questions – in order to facilitate the understanding of this complex topic to readers: What are the knowledge gaps and how do researchers tackle them? How do you see this area unfolding in the next 5 years? We think it would be extremely interesting for the readers.

However, we think the authors should be acknowledged for their work. In fact, they correctly addressed an important topic in renal cell carcinoma, and their discussion is well balanced.

We believe this article is suitable for publication in the journal although some revisions are needed. The main strengths of this paper are that it addresses an interesting and very timely question and provides a clear answer, with some limitations.

We suggest  the addition of some references for a matter of consistency. Moreover, the authors should better clarify some points.

Author Response

Response to Reviewer 1 Comments

 Point 1: Dear Editor, thank you so much for inviting me to revise this manuscript about renal cell carcinoma.

Liquid biopsy, based on the circulating tumor cells (CTCs) and cell-free nucleic acids has potential applications at multiple points throughout the natural course of cancer, from diagnosis to follow-up. The advantages of doing ctDNA assessment versus tissue-based genomic profile are the minimal procedural risk, the possibility to serial testing in order to monitor disease-relapse and response to therapy over time and to reduce hospitalization costs during the entire process. However, some critical issues related to ctDNA assays should be taken into consideration. The sensitivity of ctDNA assays depends on the assessment technique and genetic platforms used, on tumor-organ, stage, tumor heterogeneity, tumor clonality. The specificity is usually very high, whereas the concordance with tumor-based biopsy is generally low. In patients with renal cell carcinoma (RCC), qualitative analyses of ctDNA have been performed with interesting results regarding selective pressure from therapy, therapeutic resistance, exceptional treatment response to everolimus and mutations associated with aggressive behavior. Quantitative analyses showed variations of ccfDNA levels at different tumor stage. A plathora of recent studies has assessed this topic.

Thus, the paper addresses a current topic in this setting. The manuscript is well written and organized. Figures and tables are comprehensive and clear. The introduction explains in a clear and coherent manner the background of this study.

Response 1: We thank the Reviewer for his comments, and we appreciate the time and effort dedicated to providing valuable feedback on our manuscript.

Point 2: We suggest the following modifications:

Introduction section: although the authors correctly included important papers in this setting, we believe that the background of medical treatment for renal cell carcinoma should be further explained and some recently published studies should be cited within the introduction (PMID: 34894318;  PMID: 35193819), only for a matter of consistency. We think it might be useful to introduce the topic of this interesting paper.

Response2: Thank you for this suggestion. We cited in the introduction the recently published studies that you suggested (PMID: 34894318; PMID: 35193819).

Point 3: ·      Table 2 is too long. I would suggest to split it in different parts to help readability.

Response 3: We agree with this criticism, and we have, accordingly, modified Table 2 and the structure of our paper to help the readability.

Point 4: ·      The authors should expand some sections, including a more personal perspective to reflect on. For example, they could answer the following questions – in order to facilitate the understanding of this complex topic to readers: What are the knowledge gaps and how do researchers tackle them? How do you see this area unfolding in the next 5 years? We think it would be extremely interesting for the readers. However, we think the authors should be acknowledged for their work. In fact, they correctly addressed an important topic in renal cell carcinoma, and their discussion is well balanced.

We believe this article is suitable for publication in the journal although some revisions are needed. The main strengths of this paper are that it addresses an interesting and very timely question and provides a clear answer, with some limitations.

We suggest the addition of some references for a matter of consistency. Moreover, the authors should better clarify some points.

Response 4: We appreciate your input and on the base of your suggestions, we addressed these recommendations in the conclusions.

Reviewer 2 Report

Circulating cell-free DNA in Renal Cell Carcinoma: the new era of precision medicine.

Francini et al

general remarks

A good overview on the state of the art re. the use of circulating nucleic acids in plasma and serum.

Nevertheless the question arises why urine as a liquid very close to the tumor was not mentioned in a more comprehensive way.

specific comments

Page 2, line 74

blood drawing is for sure less invasive than a regular tissue biopsy but nevertheless an invasive procedure, the only exception is urine, pls correct

page 3, line 96

which papers did the authors see as not relevant and why?

Chapter 4.1.4 cfDNA epigenetic alterations

as far as I can see all cited studies in this chapter used healthy controls to be compared with RCC patients, the main challenge is the differentiation between patients with a non-malignant kidney disease (like Papillary renal adenoma , Oncocytoma or Angiomyolipoma) and a patient with cancer,

are there any data on these patient populations? If yes they need to be included in the paper

Legend of Fig 1

typo (line 450)

Author Response

Response to Reviewer 2 Comments

Point 1: Circulating cell-free DNA in Renal Cell Carcinoma: the new era of precision medicine. Francini et al

general remarks A good overview on the state of the art re. the use of circulating nucleic acids in plasma and serum. Nevertheless, the question arises why urine as a liquid very close to the tumor was not mentioned in a more comprehensive way.

Response 1: We are grateful to the Reviewer for their positive reception of our Review manuscript and for their critical comments on it. We agree with the Reviewer that urines constitute a valuable source for biomarker testing in GU oncology, due to the physical proximity of the biological fluid to the target organs, the non-invasiveness of collection and the patient compliance to be enrolled in the study. However, we wanted to focus our Review on the circulating nucleic acids measured in plasma and serum because more data are currently available and the use of these biofluids is more established in clinical practice compared to urine. Nevertheless, we agree with the Reviewer that urine-based biomarkers are of great interest and of increasing scrutiny from the scientific community, and they will be a topic for a future manuscript from our group.

Point 2: specific comments

Page 2, line 74 Blood drawing is for sure less invasive than a regular tissue biopsy but nevertheless an invasive procedure, the only exception is urine, pls correct

Response 2: Thank you for pointing this out. We agree with this comment. In fact, we modified the sentences on page 3 line 74 “Further, the analysis of cfDNA was shown to be a viable, inexpensive, and less-invasive method to diagnose and monitor cancer hence circumventing the well-known shortcomings of invasive tissue biopsies”

Point 3:page 3, line 96 which papers did the authors see as not relevant and why?

Response 3:We thank the Reviewer for this comment. As pointed out, we did not conduct a systematic review, as we aimed to provide a narrative, easy-to-read summary and overview of the cfDNA in RCC for the practicing oncologist and non-experts in the field, including the most important papers for reference and introduction. In general, we excluded the papers with the following features: non-peer reviewed, small group of patients, pediatric population, published in a very low impact journal, and published in a language other than English. Although non-systematic, we are strongly convinced that our manuscript provides the necessary, rigorous background to the readership to understand the evolving landscape of cfDNA, and we are sure that the critical comments made by the Reviewers have greatly increased the impact of our paper.

Point 4: Chapter 4.1.4 cfDNA epigenetic alterations as far as I can see all cited studies in this chapter used healthy controls to be compared with RCC patients, the main challenge is the differentiation between patients with a non-malignant kidney disease (like Papillary renal adenoma , Oncocytoma or Angiomyolipoma) and a patient with cancer, are there any data on these patient populations? If yes they need to be included in the paper

Response 4: We thank the reviewer for bringing up this important point and we agree that the main challenge is the differentiation between patients with a non-malignant kidney disease (like Papillary renal adenoma , Oncocytoma or Angiomyolipoma) and a patient with cancer. The limited data available were added at the end of the paragraph 4.1.4 line 352: “ Although they included RCC patients with  clear cell and papillary histology, no difference in methylation were found between the histology, mainly because of the small number of patients with papillary histology.”

Point 5: Legend of Fig 1 typo (line 450)

Response 5: We thank the reviewer to pointed out the error and we apologize for this typing mistake. We edited the legend accordingly: we edited “higlited for each tecnichuqe” in “highlighted for each technique”

Reviewer 3 Report

The paper is written well with good figures and tables to make the readers comfortable with the topic. I have two recommendations -

1. Instead of keeping the titles concerning cfDNA - kindly change the titles to indicate the conclusion of the following result. That way, there will be continuity to the whole narrative.

2. The authors state that it is challenging to employ the method of cfDNA for detecting RCC due to the nature of cancer. This statement diminishes the importance of work. Instead, add statements that can strengthen the reasoning - may be the success of cfDNA screening for detecting early cases for other cancers and those which are great for starting treatment on time.

Author Response

Response to Reviewer 3 Comments

Point 1: The paper is written well with good figures and tables to make the readers comfortable with the topic.

Response 1: We would like to thank you for the comments and we appreciate the time and effort that you have dedicated to providing valuable feedback on our manuscript.

Point 2: I have two recommendations -

  1. Instead of keeping the titles concerning cfDNA - kindly change the titles to indicate the conclusion of the following result. That way, there will be continuity to the whole narrative.

Response 2: Thank you for this suggestion. We modified the titles accordingly.

Point 3: 2. The authors state that it is challenging to employ the method of cfDNA for detecting RCC due to the nature of cancer. This statement diminishes the importance of work. Instead, add statements that can strengthen the reasoning - may be the success of cfDNA screening for detecting early cases for other cancers and those which are great for starting treatment on time.

Response 3: We appreciate your input and on the base of your suggestions we addressed these recommendations in the conclusion

Round 2

Reviewer 1 Report

acceptance.